# Graphene-Based Nanofluids: Production Parameter Effects on Thermophysical Properties and Dispersion Stability

**DOI:** 10.3390/nano12030357

**Published:** 2022-01-22

**Authors:** Naser Ali

**Affiliations:** Nanotechnology and Advanced Materials Program, Energy and Building Research Center, Kuwait Institute for Scientific Research, Safat 13109, Kuwait; nmali@kisr.edu.kw

**Keywords:** density, graphene, specific heat capacity, suspension thermal conductivity, viscosity

## Abstract

In this study, the thermophysical properties and dispersion stability of graphene-based nanofluids were investigated. This was conducted to determine the influence of fabrication temperature, nanomaterial concentration, and surfactant ratio on the suspension effective properties and stability condition. First, the nanopowder was characterized in terms of crystalline structure and size, morphology, and elemental content. Next, the suspensions were produced at 10 °C to 70 °C using different concentrations of surfactants and nanomaterials. Then, the thermophysical properties and physical stability of the nanofluids were determined. The density of the prepared nanofluids was found to be higher than their base fluid, but this property showed a decrease with the increase in fabrication temperature. Moreover, the specific heat capacity showed very high sensitivity toward the graphene and surfactant concentrations, where 28.12% reduction in the property was achieved. Furthermore, the preparation temperature was shown to be the primary parameter that effects the nanofluid viscosity and thermal conductivity, causing a maximum reduction of ~4.9% in viscosity and ~125.72% increase in thermal conductivity. As for the surfactant, using low concentration demonstrated a short-term stabilization capability, whereas a 1:1 weight ratio of graphene to surfactant and higher caused the dispersion to be physically stable for 45 consecutive days. The findings of this work are believed to be beneficial for further research investigations on thermal applications of moderate temperatures.

## 1. Introduction

For many years, researchers have focused on continuously enhancing the thermal performance of heat exchangers (HEs) to reduce their overall consumption of energy as well as minimize the size of their designs. This is because these systems have major roles in our everyday lives, where their uses have ranged from domestic boilers up to industrial scale power generation [1]. Historically, most attempts to improve HEs have fallen into one of two categories: geometric modifications such as regenerators, plate exchangers, and extended surfaces (e.g., addition of fins) [2], and varying the operational conditions of the working fluid by changing the fluid flow patterns (e.g., parallel and crossflow, and single- or multi-pass arrangements) [3]. The aforementioned has initially caused a substantial improvement in the performance of these devices. However, the level of enhancement in the thermal performance from adopting such approaches has reached a point where a new direction is essentially required to take the current scientific achievements to the next levels. One of the most successful approaches to accomplish this is through investigating the use of advanced working fluids than those conventionally used [2]. The primary rule is that these advanced fluids should possess outstanding thermal properties. Since nanofluids, which were introduced in 1993 by Masuda et al. [4] and named in 1995 by Choi and Eastman [5], contain favorable thermal conductivity in comparison with conventional liquids, they have the potential to be used as a replacement to the currently used working liquids to largely improve the thermal effectiveness of existing thermal exchanging systems. Generally, nanofluids are suspensions made of dispersed nanoscaled particles in a conventional hosting fluid [6]. As such, when the dispersed nanoparticles used have higher thermal conductivity compared to the host media, the effective (or overall) thermal property of their mixture will subsequently overcome that of the base fluid used. Some of the common nanoparticles used to form nanofluids include carbon-based (e.g., carbon nanotubes (CNTs), carbon black, graphene, and diamonds), metallic (e.g., iron and copper), oxides (e.g., alumina and titania), and alloys such as steel [7]. On the other hand, the commonly used base fluid is normally water, but other liquids can also be used such as oils, methanol, or a mixture between them [8,9]. It should be noted that, of the commonly used nanomaterials for producing nanofluids, CNTs, graphene, and nanodiamonds possess the highest values of thermal conductivities among the currently known nanomaterials [10,11,12,13], as shown in Figure 1. As such, the scientific community has given large attention to these types of materials in their nanofluid thermal-based investigations [14]. Some have also introduced these types of suspensions to heat transfer systems containing structures made of phase changing materials (PCM), even fabricating nanofluids out of nanocomposites that are made of, for example, graphene with PCM embedded into the pores [15]. When it comes to graphene-based nanofluids, researchers have employed it as a working fluid to enhance oil recovery [16,17], cooling hybrid photovoltaic thermal collectors [18], solar harvesting [19], and many more [20].

Even though these advanced suspensions may sound promising, there exist some factors such as the effect of the production approach on the thermophysical properties and dispersion stability, which must be considered before commercially presenting these advanced fluids into industry [22]. In addition, the higher viscosity of nanofluids compared to that of the base fluid also needs to be taken into account, especially when these suspensions are targeted toward applications that utilize them in their dynamic status. This is mainly because the pumping power requirement will eventually increase with the increase in the working fluid viscosity, and vice versa [23]. Obviously, the greater the concentration of nanoparticles, the greater the increase in thermal conductivity, but this would also result in increasing the viscosity of the suspension. As for the physical stability of the dispersed particles, it has a direct effect on the produced nanofluid thermophysical properties, in particular, the overall thermal conductivity and viscosity [7]. For example, when the nanomaterials are homogenously dispersed and physically stable in the hosting base fluid, the mixture will usually reach its optimum thermal conductivity and would have the lowest possible increase in viscosity, whereas the unstable status would have the opposite behavior. Due to the previous, it is important to have a good understanding of the factors that influence the stability of the dispersion of nanoparticles in the nanofluid. Factors to be considered include the production process (single-step or two-step approach), duration of the mixing process, temperature at which the dispersion process is carried out, the base liquid used, molecular forces between the nanoparticles used and the molecules of the base liquid, concentration of the nanoparticles, and the type and concentration of the surfactant used (if any) [24,25]. Changes in any of these factors will result in changes in the outcome of the thermophysical properties of the nanofluid, even if the same base liquid and nanoparticles are used. On the molecule scale, researchers have shown that the diffusive heat conduction and Brownian motion of the nanoparticles have some effect on the thermal conductivity of nanofluids [26]. Furthermore, the thermal conductivity and viscosity of a suspension can accurately be obtained through experimental measurements (Figure 2) [27,28,29,30]. However, theoretical correlations also exist with relatively acceptable prediction capabilities [31,32,33,34,35,36,37].

It must be noted that from the different thermal conductivity characterization devices that are listed in Figure 2, the transient hot-wire method is the widely used approach to obtain this thermal property for these types of dispersions [13,38,39]. The reason behind this relies on its capability of eliminating measurement errors that can be present from the natural convection behavior developed with the fluid. In addition, it offers a prompt measuring response of a couple of seconds and the conceptual design of the device is considered simple in comparison with other devices made for similar purposes. In terms of the thermo-economical aspect, scholars have found that the feasibility of using nanofluids depends mainly on the targeted application. However, more focus is needed toward such type of studies [40,41].

There are many available literatures on the viscosity and thermal conductivity of carbon-based nanofluids. For example, Aravind et al. [42] studied the changes in viscosity of four groups of carbon nanotube (CNT)-based nanofluids at relatively low volume percentage (vol. %) of concentrations, precisely 0.005 vol. % and 0.03 vol. %, using water and ethylene glycol (EG) as the base fluids, respectively. The scholars measured the viscosity of their as-prepared nanofluids at various temperatures (i.e., 30 °C to 70 °C) where they found that this thermal factor has a dominant impact on the property. Halelfadl et al. [43] investigated the effect of CNT concentration and suspension temperature on the viscosity of CNT-based dispersions. They found that raising the nanomaterial concentration caused the viscosity to increase due to the higher aggregates of CNTs formed. However, the rise in temperature demonstrated a reduction in the viscosity of their as-prepared nanofluids. In addition, the study performed by Ding et al. [44] also confirmed the previous conclusion of Halelfadl et al. [43]. In terms of the effect of preparation parameters on the thermal conductivity of CNT-based nanofluids, Ding et al. [44] found that the thermal property could increase by 10% over the base fluid by dispersing 0.49 vol. % of the carbon-based nanomaterial. However, this level of improvement in the thermal conductivity could be further enhanced to 79% by increasing the produced suspension temperature from 20 °C to 30 °C at the same nanomaterial concentration. The previous work published by Yang et al. [45] also support the previous findings, where they attribute this enhancement in nanofluid thermal conductivity toward the increase in Brownian motion of the dispersed nanoparticles as the result of the rise in temperature. Furthermore, the review conducted by Murshed and Castro [46] showed that the thermal conductivity of CNT-based nanofluids increased more pronouncedly with temperature compared to the concentration. Yarmand et al. [47] studied the thermal property of graphene nanoplatelets-based nanofluids. The authors used water as their base fluid and 0.02–0.1 wt. % of the carbon-based nanomaterial to form their suspensions after the nanopowder was functionalized through the acidic treatment of the H_2_SO_4_ and HNO_3_ mixture (1:3 weight ratio). Furthermore, the as-prepared powder was then dispersed at 20–40 °C through the two-step nanofluid production route. The scholars discovered that the production temperature had a strong effect on the as-produced nanofluid thermal property as well as the concentration of the as-prepared graphene that was employed. In addition, the highest enhancement recorded was 19.68% over that of the base fluid when using 0.1 wt. % concentration and a 40 °C preparation temperature. Ghozatloo et al. [48] investigated the influence of dispersed nanomaterial concentration, preparation temperature, and time on the thermal conductivity of graphene–water suspensions. The researchers used two groups of graphene, namely, pure and functionalized graphene. They then produced their nanofluids through ultrasonicating the carbon-based nanomaterials with the base fluid with concentrations of 0.01–0.05 wt. %. It was found that the pure graphene-based nanofluids promptly lost their physical stability, whereas the functionalized group had a high level of physical stability. Moreover, the thermal conductivity of the functionalized group demonstrated an enhancement of 17% (at 50 °C and 0.03 wt. %) and 13.5% (at 25 °C and 0.05 wt. %) over the base fluid at the same temperature condition. Askari et al. [49] studied the effect of the surfactant on the thermal conductivity of graphene-based nanofluids. They used water from South Iranian cooling towers as their bas fluid, after which they dispersed graphene (0.1–1.0 wt. %) with different surfactants, namely gum Arabic, Tween 80, Triton X-100, acumer terpolymer, and cetrimonium bromide. It is important to note that the authors used these types of base fluids to reflect the actual case scenario for the possibility of future implementation of these advanced working fluids. Furthermore, the scholars used both zeta potential analysis as well as image capturing technique to determine the stability of their suspensions. They found that Tween 80, as a surfactant, provided a highly stable nanofluid that lasted for about 2 months. Moreover, the level of enhancement in the thermal conductivity of their as-prepared nanofluids with Tween 80 reached 16% (at 45 °C and 1.0 wt. %) over that of the base fluid. The authors also noted that using a low nanomaterial concentration is preferable for industrial applications, since they would have less impact on the viscosity and density of the working fluid, which would consequently result in less pumping power requirement.

In light of the previous literature survey and other published work [50,51,52], this research studied the effect of production parameters on the thermophysical properties and physical stability of carbon-based dispersions. Graphene nanoplatelet nanopowder was used at a concentration of 0.01–0.1 vol. % to form the suspensions. In addition, sodium dodecyl sulfate (SDS) surfactant was added with a weight ratio of 0.5–1.5:1 (SDS to graphene) in order to improve the dispersion physical stability. This is because graphene is physically unstable in water due to the π-conjugative structure of the nanomaterial and high hydrophobicity of its surface [53], and thus the use of a surfactant is essential in the production process to improve the stability of such types of suspensions [54,55]. The two-step controlled production temperature method was employed to disperse the nanomaterial. This was conducted by ultrasonicating the mixture for a total duration of 1.5 h while controlling the temperature of the water bath from 10 °C to 70 °C with a probe type ultrasonicator. After the production phase, the density and specific heat capacity of the suspensions were determined. In addition, the effective thermal conductivity and viscosity of the as-produced suspension were experimentally determined with a hot-wire instrument and a viscometer device, respectively. These measured thermophysical properties were then compared to those obtained for the base fluid to assess the degree of change in the properties. Furthermore, the short- and long-term physical stability was also determined using the UV–Vis approach and image capturing approaches for 45 days. To the best of the authors’ knowledge, there is no previous work that has focused on determining the physical stability as well as all the thermophysical properties of graphene nanoplatelet–water nanofluids using the previous experimental setup and conditions. Since these types of suspensions are very sensitive to temperature, nanomaterial concentration, and the type as well as the amount of surfactant used, the outcome of this work is believed to be beneficial for interested scholars in simulating studies that include graphene–water suspensions in moderate temperature applications such as gas turbine intercooling units [22], liquid cooled computers [56], low temperature parabolic trough solar collectors [57,58], and other thermal energy systems [59]. It will also provide an insight into future experimental work for these applications.

## 2. Experimental Procedure

### 2.1. Materials and Equipment

Commercial graphene nanoplatelet powder with a purity of >99.5 wt. %, thickness of 2 to 8 nm, diameter of 4 to 12 µm, and formed with an average of three to six layers was supplied by from US Research Nanomaterials Inc. (Houston, TX, USA). Sodium dodecyl sulfate (SDS), of type ReagentPlus^®^ and with ≥98.5% purity, was obtained from SIGMA-ALDRICH Inc. (St. Louis, MO, USA) and then added as a surfactant to the suspensions at different concentrations. Clear glass vials of 100 mL volume, 1.6 mm wall thickness, 40 mm outer diameter, and 95 mm height were purchased from Glass Solutions Limited (Hertfordshire, UK). A paraffin and polyolefin wax-based thin film was obtained from PARAFILM^®^ M (Neenah, WI, USA). This wax film will be used later to seal the vials throughout the nanofluid dispersion process to prevent volumetric losses of the base fluid due to the evaporation mechanism. Figure 3 shows the nanomaterial, surfactant, wax film, and vial that were used in the experiment. Deionized water was produced from an Elga PR030BPM1-US Purelab Prima 30 water purification system (Buckinghamshire, UK), after which it was used as the base fluid for the suspension following neutralizing its pH (i.e., pH of 7 at a 20 °C surrounding). The adjustment of the pH value was conducted by adding sodium hydroxide solution (+pH), of type 1.09956. Titrisol^®^ and hydrochloric acid ~37% (−pH) were of ACS reagent grade. Both chemicals used for adjusting the pH value were obtained from SIGMA-ALDRICH Inc. (St. Louis, MO, USA), and while the pH was modified, the liquid was monitored with a PHC20101 Intellical gel filled Ph electrode (Loveland, CO, USA) connected to a HACH HQ11D portable pH meter (Loveland, CO, USA) with a ±0.002 pH accuracy. Furthermore, the calibration of the pH meter was conducted using pH 10, 7, and 4 commercial calibration fluids that were purchased from Metrohm USA Inc. (Fountain Valley, CA, USA).

### 2.2. Powder Analysis

Elemental examination was conducted on the graphene feedstock using a 9 kW X-ray diffraction (XRD) analyzer (Rigaku SmartLab, Tokyo, Japan). The utilized XRD device runs on commercial software, SmartLab Guidance, which was supplied with the system and a CuK_α_ source type of X-ray as well as a 2θ diffraction angle. Furthermore, the incidence beam step used was 0.1° with a diffraction scanning angle in the range of 10° to 80°, and 1°/min scanning rate. The aforementioned parameters were used to determine the Bragg’s peaks in the analyzed specimen’s elemental content. Moreover, the morphology of the as-received feedstock nanopowder as well as the impurities were explored using a field emission scanning electron microscopy (FE-SEM) system, of type JEOL JSM-IT700HR (Tokyo, Japan), along with its integrated energy dispersive X-ray spectroscopy (EDS) unit, and InTouchScope (ver. 1.12) supplied software. The FE-SEM images were recorded at two different magnifications by the secondary electron mode from the surface region of the tested sample. It should be noted that the powder sample was initially coated with gold (Au) to increase its conductivity; the working distance in which both FE-SEM and EDS analyses were performed was 12 mm above the examined sample; and a 5 kV accelerating voltage was employed to lower the potential damage to the nanomaterial. In terms of the density, this property was determined for the graphene nanoplatelets so that it can be used later to determine the required amount of nanomaterial based on the planned nanofluid vol. %. As such, 3 g of the nanopowder was measured using an analytical balance of type ae-ADAM PW 214 (Oxford, CT, USA) with accuracy: ±0.2 mg, and readability: 0.1 mg). The as-measured graphene nanoplatelets were then placed carefully in the sample holder located inside the gas pycnometer-volumetric device of type HumiPyc trademark (Model 1). The device was afterward operated at 20 °C to measure the allocated graphene nanoplatelets volume ($V_{GN}$), then we calculated its density ($\rho_{GN}$) from the initially known powder mass ($m_{GN}$) through the following equation:(1)$$\rho_{GN}=\frac{m_{GN}\;}{V_{GN}}$$

The device output showed that the value of $\rho_{GN}$ was 127 kg/m^3^.

### 2.3. Base Fluid Property Analysis

The thermophysical properties, which include the density, specific heat capacity, thermal conductivity, and viscosity, were determined for the base fluid through experimental means in a 10–70 °C range of controlled temperature. For the density measurement, a density meter instrument of type DMA 4500M (Anton-Paar Co., Washington County, CO, USA) was employed. The device has an automatic cleaning and calibration feature and utilizes 20 mL of the examined liquid to determine its density at the required targeted temperature. For the specific heat capacity, a DTA/DSC device, of type LABSYS evo (SETRAM Instrumentation Co., Caluire-et-Cuire, France), was used. This system requires the user to include a small amount of the liquid, using a pipette type dropper, within the sample holder to the reference height. Next, the device was activated with the inputted set of controlled temperatures to obtain the property. Furthermore, the base fluid thermal conductivity was determined with a hot-wire instrument of type THW-L2 (Thermtest Co., Hanwell, NB, Canada). The measurements were conducted by first placing the glass vial on a hot/cold plate (Thermtest Co., Hanwell, NB, Canada), then 75 mL of the base fluid was added, after which the temperature of the thermal source was set to the required controlled temperature (i.e., 10–70 °C). Following the previous step, the THW-L2 probe was immersed in the liquid and three measurements of the thermal property were taken, with a duration of 5 min between them, then the values were averaged. On the other hand, the viscosity of the base fluid was obtained with a viscometer device of type RV-2T (provided from Brookfield Co., Toronto, ON, Canada) at the same range of temperatures that were used with the previous thermophysical property tests. Initially, the liquid was placed in the vial, then its temperature was controlled in a similar manner to that used with the thermal conductivity tests. Afterward, the viscometer spindle was inserted into the base fluid to its reference point (i.e., a thin line placed by the manufacturer). The device was then set to its scanning mode to obtain the rpm value best suited to analyzing the examined sample, which was afterward employed to determine the value of the property at each controlled temperature.

### 2.4. Dispersion Fabrication

SDS surfactant was added to the base fluid at different weight ratio of SDS:graphene (i.e., 0.5–1.5:1) then stirred for 10 min using a magnetic stirrer to ensure that the powder was fully dissolved. Then, graphene powder was inserted in a vial, then 75 mL of the previously prepared base fluid was injected above the nanomaterial. Afterward, a Sonics Materials VCX 750 ultrasonic microprocessor, which is a probe type sonicator obtained from Fisher Scientific Co. (Portsmouth, NH, USA), was used to disperse the nanoparticles within the base fluid for 1.5 h while tightly sealing the vial opening with the sealing film. It is important to note that the ultrasonic device operated at 750 W net power output and 20 kHz with a 19 mm solid probe made of a high-grade titanium alloy (Ti–6Al–4V). Furthermore, the mixture temperature was controlled (i.e., 10–70 °C) throughout the dispersion process using a BUENO BIOTECH company cooling and heating water bath, of type BGDC, with a 0.1 °C accuracy while being monitored using the ultrasonic temperature probe accessory, which is made of stainless steel and has a 1 °C accuracy. The previous fabrication process is based on the well-known two-step nanofluid production approach [7,60], and was adopted for the formation of the nanofluid samples of different concentrations (i.e., 0.01–0.10 vol. %) at a lab temperature of 25 °C. Furthermore, the calculation of the required vol. % of nanomaterial toward base fluid volume ($V_{bf}$) used (i.e., 75 mL) was conducted using the mixing theory [7,61], as shown in Equations (2) and (3).
(2)$$vol.\%=\frac{V_{GN}\;}{V_{GN}+V_{bf}}\times 100;$$
(3)$$vol.\%=\frac{\frac{m_{GN}\;}{\rho_{GN}}\;}{\frac{m_{GN}\;}{\rho_{GN}}+V_{bf}}\times 100;$$

### 2.5. Suspension Thermophysical Property and Physical Stability

Thermophysical property of the as-prepared graphene-based nanofluid was determined using a similar approach as the one previously used for the base fluid. As such, the values of the thermal conductivity, specific heat capacity, density, and viscosity of the dispersions were obtained. Moreover, the viscosity and thermal conductivity enhancement over the hosting fluid (i.e., water) were explored. On the other hand, the physical stability of the suspension was determined through two approaches, namely the photograph capturing route and the UV–Vis spectrum method. In the photograph capturing approach, a Canon EOS 700D professional camera (Canon Inc., Tokyo, Japan) equipped with a 105 mm micro lens of type Sigma F2.8 EX DG (Sigma Co., Fukushima, Japan) was used to capture the images of the as-prepared suspensions following their preparation and after 45 days of shelving. As for the UV–Vis analysis, a SHIMADZU UV-2600 device (SHIMADZU Co., Kyoto, Japan) was used along with its software, UVProbe ver. 2.61, to examine the as-prepared samples’ absorbance directly after production and on the 45th day. The wavelength range used was from 200 nm to 800 nm, and the sampling interval was 0.5 nm.

## 3. Results and Discussion

### 3.1. X-ray Diffraction Characterization

In the XRD characterization conducted on the obtained commercial graphene nanoplatelets, the electromagnetic beam that generates from the X-ray source is reflected from the crystalline plane. The reflection angle obtained corresponds to the crystalline structure of the tested sample. When comparing the outcome of the reflected beam from the different crystalline planes to the device database, the spectra of the nanomaterial can be revealed for the examined powder. Furthermore, comparing the generated diffraction pattern from the conducted characterization (Figure 4) to the available literature (e.g., Jiang et al. [62] and Prolongo et al. [63]), it can be seen that the results acquired were almost similar. The previous concluded that the examined nanopowder was graphene nanoplatelets. This was also confirmed from the device database (i.e., PDF# 04-014-0362). In addition, the highest peak (0 0 2) shown in the XRD pattern corresponded to the graphene sheets stacking together [64], and therefore supports the manufacturer claims of providing multi-layered graphene nanoplatelets (i.e., of three to six layers). It should be noted that the XRD patterns of carbon nanotubes and graphite are relatively similar to that of graphene because of the intrinsic nature of these carbon-based nanomaterials [65]. Moreover, the crystallite size of each crystalline structure was calculated using the Scherrer formula [66,67,68,69], which is demonstrated in Equation (4).
(4)$$D_{hkl}=\frac{F\lambda }{\beta_{hkl}cos\theta_{hkl}}$$
where F, $\theta_{hkl}$, $\beta_{hkl}$, and λ are the shape factor (equals ~0.9), Bragg angle at the (hkl) peak, full width at half the maximum of the (hkl) diffraction peak, and wavelength of the CuK_α_ X-ray radiation source (~0.1541 nm), respectively. At the (0 0 2) peak (i.e., the highest peak), the crystallite size was found to be 40.88 nm.

### 3.2. FE-SEM and EDS Characterization

The FE-SEM characterization of the commercially obtained graphene powder showed that the nanomaterial used has platelets with a morphological shape. It also showed that there exists some sort of stacking between these nanoplatelets, which can be attributed to the high surface energy of such nanomaterials as a result of their large surface to volume ratio. The nanoplatelets tend to attract each other to lower their surface energy, and hence achieve a more stabilized state of thermodynamic condition. Moreover, the apparent diameter was found to be between ~2 to 10 µm, which was very close to what was reported by the manufacturer (i.e., 4 to 12 µm). Such variation in diameter in commercial nanopowders is normal and has been previously experienced by Singh et al. [70]. Figure 5a,b shows the FE-SEM patterns at high and low magnifications, where the stacking of the nanoplatelets can clearly be observed. Furthermore, the elemental spectrum and mapping through the EDS analysis (Figure 5c,d) demonstrated that the nanopowder sample was made purely of carbon, which is the key element of graphene. However, the Au element seen in the EDS X-ray spectrum (i.e., Figure 5c), which was excluded from the analysis outcome, was present due to the coating material that was required in the characterization process, as explained previously in Section 2.2.

### 3.3. Base Fluid Thermophysical Properties Measurement

The water properties that were determined at 10–70 °C are demonstrated in Figure 6 and Figure 7. Furthermore, the density showed a reduction in its value with the rise in fluid temperature. The lowest and highest values recorded were 0.97843 g/cm^3^ (at 70 °C) and 0.99973 g/cm^3^ (at 10 °C), respectively. These values show good agreement to the results obtained by Baboian [71], with a variation of less than ~1.16%. In terms of specific heat capacity, the measurement results did not seem to follow a certain trend behavior with temperature changes, however, they were confined between 4.178 kJ/kg.K (lowest value) and 4.193 kJ/kg.K (highest value) at 30 °C and 10 °C, respectively. On the other hand, the thermal conductivity demonstrated a maximum increase in its value of ~13.75% with the rise in fluid temperature. The average values of the property at 70 °C and 10 °C were found to be 0.662 W/m.K and 0.582 W/m.K, respectively. It should be noted that such thermal conductive behavior of water is natural (i.e., increases when raising the fluid temperature), whereas oil and other organic liquids tend to behave oppositely, as reported by Poling and Prausnitz [72]. Furthermore, the dynamic viscosity demonstrated a high sensitivity to temperature variation of the fluid, where it showed a maximum reduction of ~69.06% when the temperature of the liquid was raised to 70 °C. Both thermal conductivity and viscosity analysis of the base fluid at controlled temperatures (i.e., 10 °C to 70 °C) can be seen in Figure 7. It should be noted that the kinematic viscosity ($\vartheta_{bf}$) can be determined at any temperature from the dynamic viscosity *(*${µ}_{bf}$) and density ($\rho_{bf}$) of the fluid at that given temperature through Equation (5)
(5)$$\vartheta_{bf}=\frac{{µ}_{bf}}{\rho_{bf}}$$

### 3.4. Suspension Thermophysical Properties Characterization

The thermophysical properties, which included the density, specific heat capacity, thermal conductivity, and viscosity, of the nanofluids were measured following the fabrication of the suspensions. It was found that the density increased with the increase in both the surfactant as well as the dispersed nanomaterial concentration, with the second having a more dominant effect on the property due to its solid nature compared to the dissolved surfactant in the examined temperature range (i.e., 10 °C to 70 °C). However, due to the low concentration of nanomaterial used in the nanofluid fabrication (i.e., 0.01–0.1 vol. %), the changes in the density value were small. For example, the 1.5:1 surfactant ratio and 0.1 vol. % of dispersed graphene, which represent the samples with the highest concentration, showed an increase in the property of ~0.17% and ~1.09% at 10 °C and 70 °C, respectively. It was also noticed that the deviation between the density values clearly widened when the high SDS ratio and graphene concentration nanofluid fabrication temperature was controlled at 40 °C and above to 70 °C, whereas the property for the other lower concentration samples at these temperatures were roughly close to that of the base fluid. It should be pointed out that, since density is defined through the amount of packed atoms within a volume, adding more surfactant and/or nanomaterial to the base fluid would cause the property to increase. However, increasing the temperature (i.e., 10 °C to 70 °C) of the dispersion would cause the bond distances between the base fluid atoms to widen, and thus the density reduces. The aforementioned has less effect on the dispersed graphene due to its solid nature and stronger atomic bonds. The variation in nanofluid density with temperature is illustrated in Figure 8. Moreover, the specific heat capacity value of the nanofluids was seen to reduce when increasing the graphene concentration as well as the surfactant ratio. Such behavior is common with these types of fluid [73,74], as increasing the dispersed nanomaterial concentration and/or adding surfactant in the base fluid (i.e., water) would lead to increasing the liquid heat transfer capability, but at the same time, would reduce its initial thermal storage capability. Figure 9 demonstrates the variation in the produced suspensions’ specific heat capacity with respect to their fabrication temperature. Further analysis of the obtained data in Figure 9 illustrates that using 0.1 vol. % of graphene nanoplatelets with a 1.5:1 surfactant ratio caused the thermal property of the base fluid to be reduced to values between ~28.12% (70 °C) and ~27.47% (10 °C). It can be concluded from such relatively high variation in the values that the specific heat capacity is highly sensitive in graphene-based nanofluids, even when low concentrations are used. In Figure 8 and Figure 9, it should be noted that the NF in the plot refers to nanofluid and the ratio between brackets refers to the SDS:graphene ratio.

In addition to the previously examined nanofluid thermophysical properties, the level of improvement in the thermal conductivity was shown to increase with the increase in graphene concentration and fabrication temperature, whereas increasing the SDS concentration caused the thermal property to decrease. This can be justified through understanding the mechanism in which the SDS dispersant enhances the dispersion stability of the suspension, since they have a tendency to form a thin layer on the outer surface of the nanomaterial to reduce its tendency to attract or cluster with other nanoplatelets [7]. This newly introduced thin layer has a much lower thermal conductivity compared to the dispersed graphene, and thus reduces the actual exposure surface of the nanoplatelet to the surrounding base fluid and consequently reduces the overall thermal conductivity of the mixture. However, the increase in thermal conductivity with the rise in temperature is attributed to the increase in Brownian motion and kinetic energy of the dispersed nanomaterial as a result of the temperature elevation [75]. Figure 10 demonstrates the level of improvement in the fabricated suspensions’ thermal conductivity compared to the base fluid. The largest level of enhancement in the thermal conductivity was ~125.72%, which was obtained using the 0.1 vol. % suspension, made at 70 °C with a dissolved surfactant ratio of 0.5:1.

In terms of the viscosity, the outcome from the measurements revealed that the increase in the property that resulted from dissolving the SDS, at the three different sets of ratios, had a negligible effect. On the other hand, increasing the dispersed graphene concentration showed a low effect on the viscosity of the suspension. This is mainly attributed to the low concentration (i.e., 0.01–0.1 vol. %) of dispersed graphene nanoplatelets employed; in many cases, researchers prefer to use a nanomaterial concentration of <1 vol. % to maintain the Newtonian behavior of the base fluid [21]. However, the fabrication temperature was found to have a dominant effect on the viscosity. For instance, the nanofluids produced with 0.1 vol. % graphene showed an increase in their viscosity over that of their base fluid by ~4.9% and ~1.38% at 10 °C and 70 °C, respectively. Figure 11 demonstrates the variation in the dynamic viscosity ratio with respect to graphene concentration and temperature. Generally, the Max. and Min. enhancement in viscosity over that of water were found to be ~4.9% (0.1 vol. % and 10 °C suspension) and ~0.28% (0.01 vol. % and 70 °C suspension), respectively.

In addition to the earlier property characterizations, the physical stability of the suspensions was examined for 45 days. The results showed that all three ratios of dissolved SDS had initially provided the nanofluids with good physical stability behavior of up to one day. However, the samples with a 0.5:1 ratio (SDS:graphene) could not maintain their long-term dispersion stability (i.e., for 45 days). This finding was also supported by the UV–Vis absorbance analysis, where the 0.5:1 samples showed very poor absorbance after 45 days of shelving compared to the other two higher SDS ratios. For example, at 30 °C, the low SDS ratio samples showed a reduction of ~75% in UV absorption capability on day 45 compared to when they were analyzed directly after their production. The aforementioned illustrates the important role that the surfactant concentration has on the suspension physical stability on both short- and long-term shelf life. Figure 12 shows the UV–Vis absorbance results and visual examples of the 0.1 vol. % as-prepared nanofluid (at 30 °C) after their production and on the 45th day. However, it should be noted that if these nanofluids were to be used in real-life applications (e.g., heat exchangers), the selection between low surfactant ratio and the other two higher ones mainly depends on the time of fabrication and implementation of these working fluids. This means that if the user is capable of producing and using their nanofluids on site without having to store the suspension, then the stability issue will be of less importance and the focus should be directed toward the thermophysical properties of the working fluid (i.e., density, specific heat capacity, thermal conductivity, and viscosity), especially since mixing does occur naturally when these fluids are under dynamic condition.

## 4. Limitations and Future Research Direction

Part of the limitations of this work is that researchers using different types of mixing devices such as a stirrer or ball-milling, or even when varying the dispersion intensity and duration, would likely obtain different thermophysical properties and physical stability. The users’ experience as well as the brand of the equipment, nanomaterial, and surfactant would also have some sort of influence on the properties and condition of the nanofluid. These factors need to be considered by interested scholars in the field.

In terms of the future research direction, it is believed that scholars should focus on characterizing the thermophysical properties and physical stability of nanofluids made of other types of carbon-based nanomaterials such as nanodiamonds and carbon black. In addition, the effect of different two-step production approaches such as ball- or roll-milling and magnetic stirring on the resulting effective thermal conductivity and dispersion stability should also be investigated. Furthermore, correlation development from experimental data will be very useful for those interested in modeling such types of advanced fluids in different applications. Investigating dispersed graphene in seawater with dissolved surfactant for solar still systems also needs to be explored. However, it is very important to include chemical analysis and elemental tests to ensure that no contamination is present in the produced water due to possible nanomaterial transportation. A feasibility study is also required and should be considered before employing these nanofluids in real-life applications. It should at least include a cost analysis, system performance evaluation, and the environmental impact from utilizing these suspensions.

## 5. Conclusions

In this study, the thermophysical properties and physical stability of graphene–water nanofluids were investigated through different controlled production parameters. The reason behind this investigation was to provide insights into these types of advanced fluids for potential use in moderate temperature applications. Furthermore, the main parameters that were explored are the controlled temperatures at the mixing stage, dispersed nanomaterial concentrations, and added surfactant weight ratios. A two-step controlled temperature method from 10 °C to 70 °C was used to fabricate the suspensions of 0.01 to 0.10 vol. %. In addition, the SDS surfactant was used to enhance the physical stability of the as-prepared carbon-based nanofluids at different sets of weight ratios, precisely 0.5–1.5:1 SDS to graphene wt. %. The following conclusions were drawn from the conducted research.

Adding a surfactant (i.e., SDS) increases the physical stability of graphene-based suspensions. However, it lowers their thermal conductivity and specific heat capacity.The density of the base fluid increases with the increase in disperse particle concentration as well as the surfactant weight ratio. On the other hand, the property decreases with the rise in base fluid/nanofluid temperature.The specific heat capacity was shown to be very sensitive to both surfactant and nanomaterial concentrations, where increasing the concentration of either of these factors resulted in a reduction in the property. However, the production temperature was shown to have significantly less effect on the specific heat capacity of the suspension.The thermal conductivity of the suspension was found to be highly influenced by the production temperature compared to the concentrations of the dispersed graphene nanoplatelets and surfactant, which demonstrated that these two factors have less impact on the property.On the other hand, the viscosity was found to increase with the increase in nanomaterial concentration and reduce with the increase in production temperature. In addition, the surfactant concentration showed no effect on the property, which is believed to be attributed to the low concentration used in the investigation.The physical stability analysis showed that the low surfactant weight ratio (i.e., 0.5:1 of SDS to graphene wt. %) could maintain a good dispersion stability for 24 hours. However, using higher surfactant weight ratios can extend the physical stability of the graphene–water nanofluids for 45 days.

## Figures and Tables

**Figure 1 nanomaterials-12-00357-f001:**
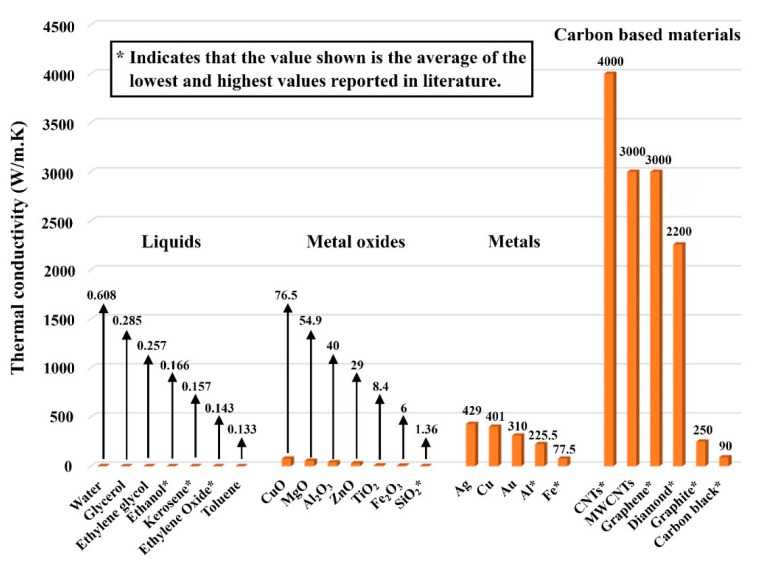
Thermal conductivity of some of the common base fluids and solid feedstocks used in fabricating nanofluids [21].

**Figure 2 nanomaterials-12-00357-f002:**
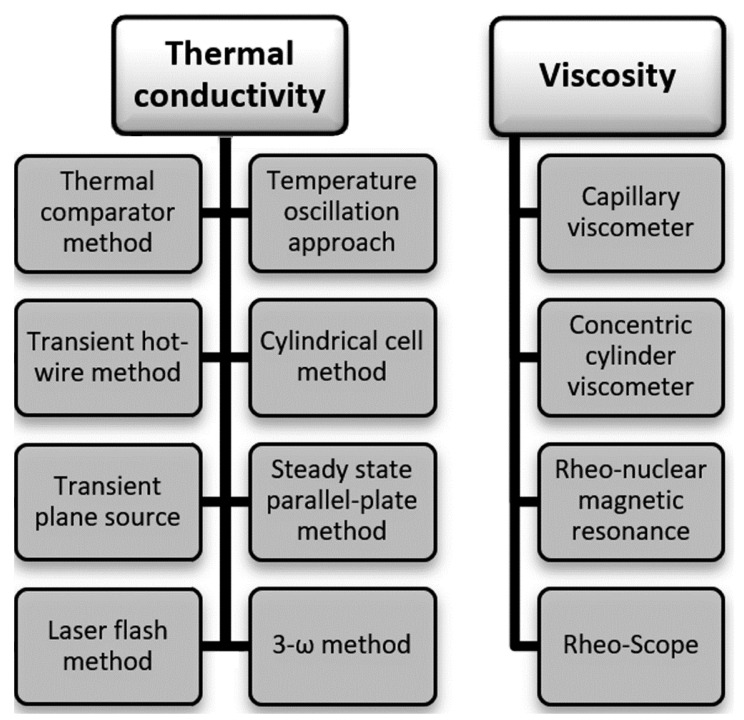
List of devices used to determine the thermal conductivity and viscosity.

**Figure 3 nanomaterials-12-00357-f003:**
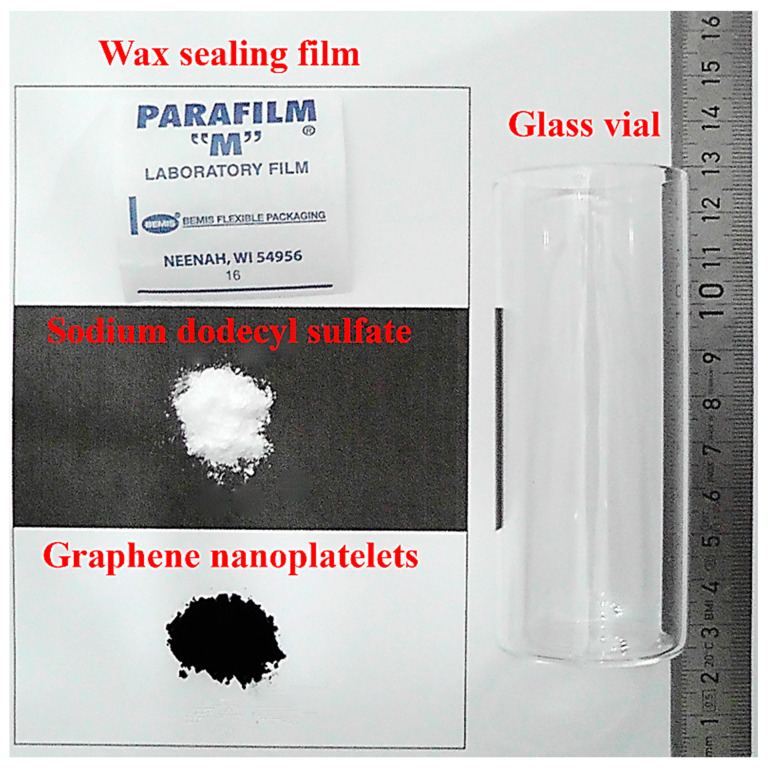
Experimental materials used in the nanofluid preparation process, which include the glass vial, wax film, SDS surfactant, and graphene nanoplatelet nanopowder.

**Figure 4 nanomaterials-12-00357-f004:**
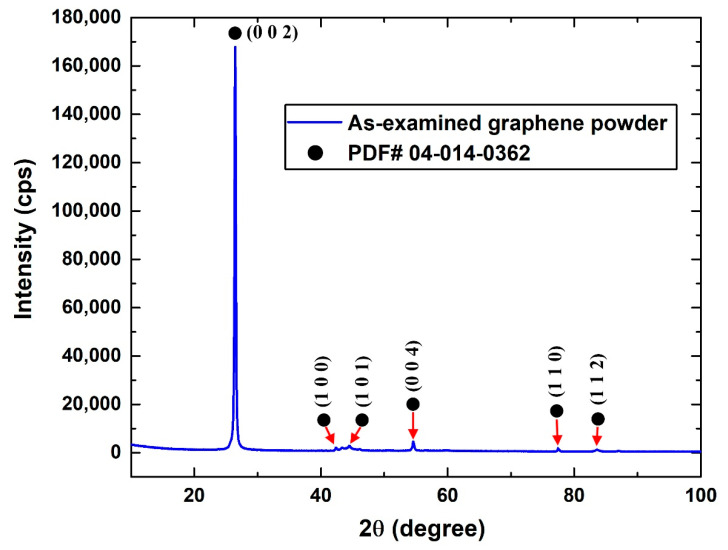
X-ray diffraction analysis of the supplied graphene nanoplatelets.

**Figure 5 nanomaterials-12-00357-f005:**
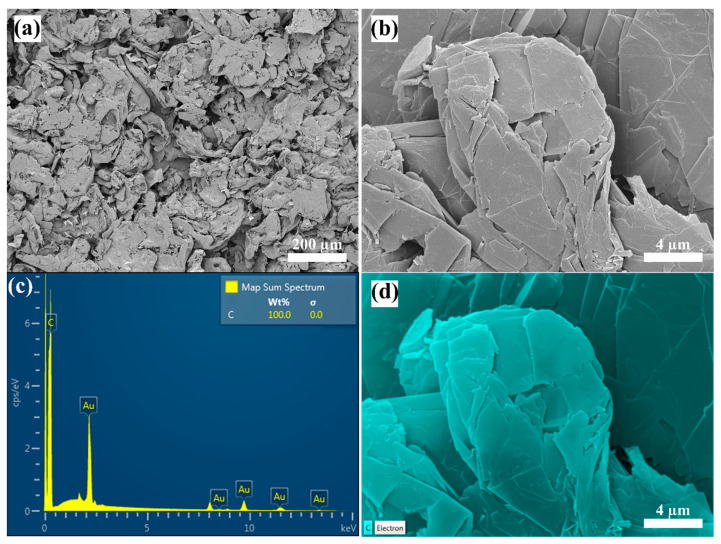
FE-SEM and EDS analysis, where (**a**,**b**) demonstrates the morphology at low and high magnifications, and (**c**,**d**) shows the X-ray spectrum and elemental mapping, respectively.

**Figure 6 nanomaterials-12-00357-f006:**
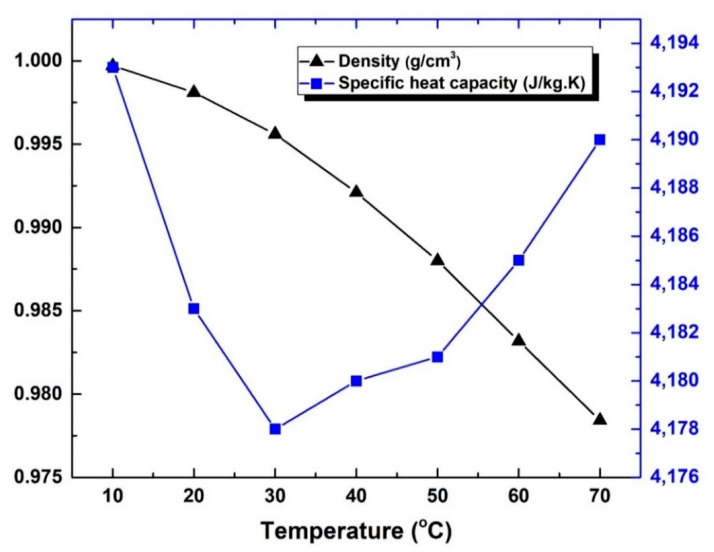
Variation in the water density and specific heat capacity at 10–70 °C.

**Figure 7 nanomaterials-12-00357-f007:**
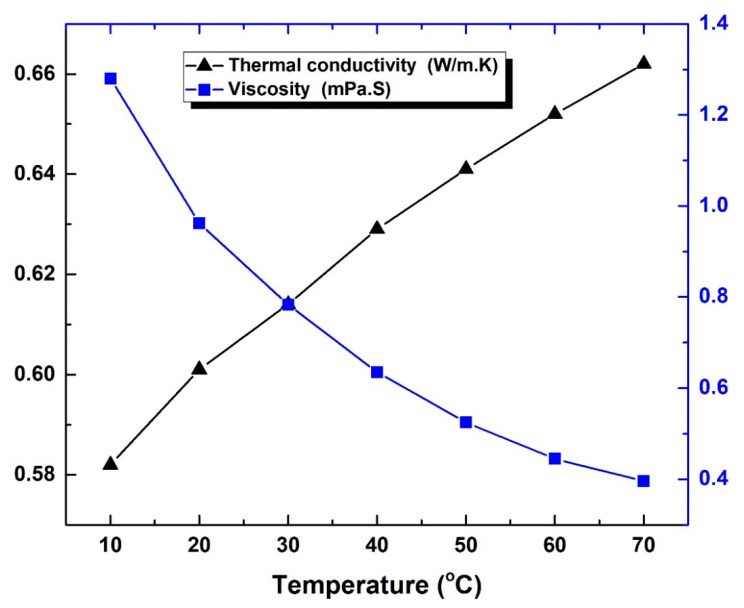
Variation in the water dynamic viscosity and thermal conductivity at 10–70 °C.

**Figure 8 nanomaterials-12-00357-f008:**
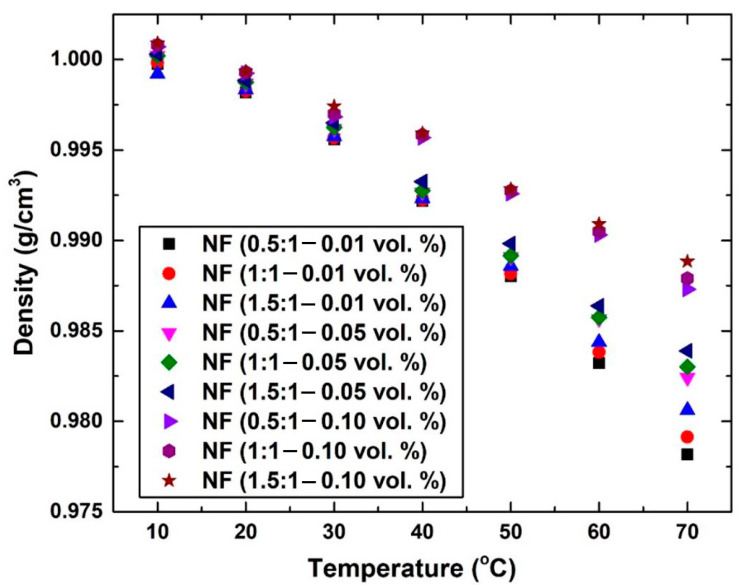
Nanofluids’ density variation at different controlled temperature, surfactant ratio, and graphene concentration.

**Figure 9 nanomaterials-12-00357-f009:**
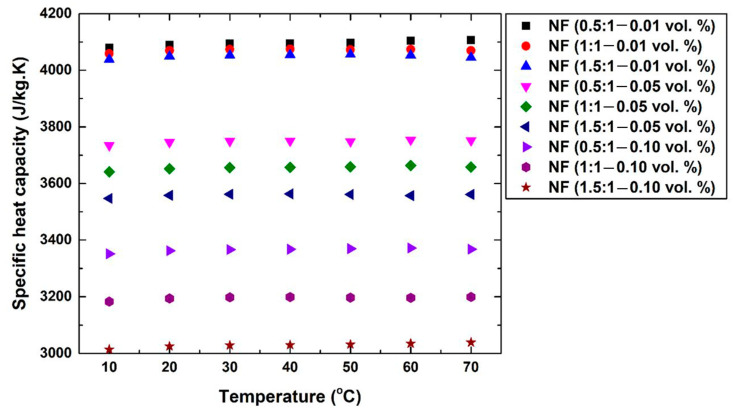
Nanofluids’ specific heat capacity variation at different controlled temperature, surfactant ratio, and graphene concentration.

**Figure 10 nanomaterials-12-00357-f010:**
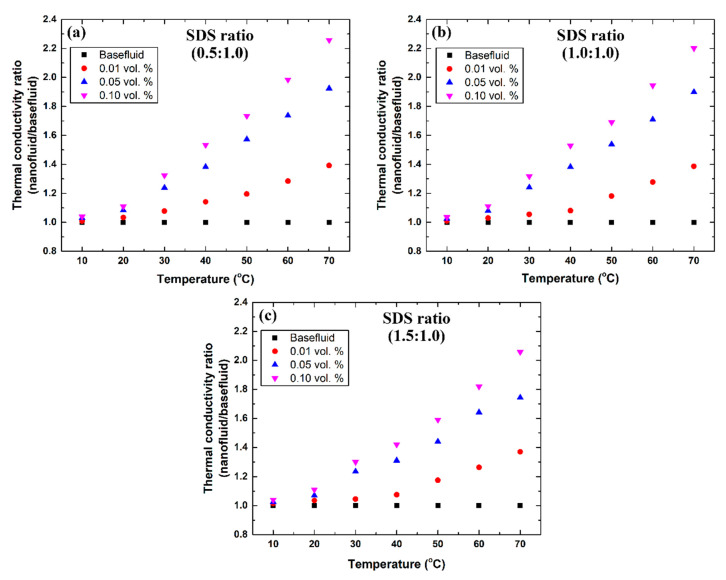
Variation in the thermal conductivity ratio of the suspension to the base fluid at different temperatures and surfactant concentrations, where (**a**–**c**) shows the samples made with the 0.5:1, 1:1, and 1.5:1 ratio of SDS to graphene, respectively.

**Figure 11 nanomaterials-12-00357-f011:**
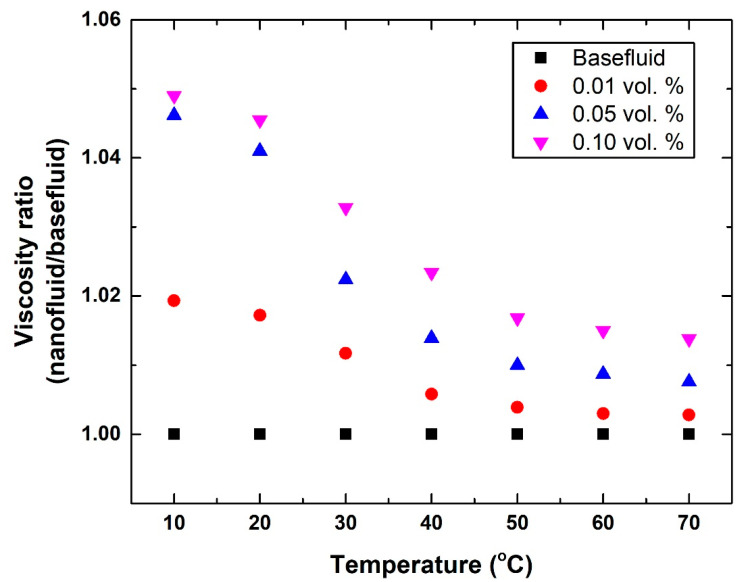
Variation in the viscosity ratio of the suspension to the base fluid with respect to graphene concentration and fabrication temperature.

**Figure 12 nanomaterials-12-00357-f012:**
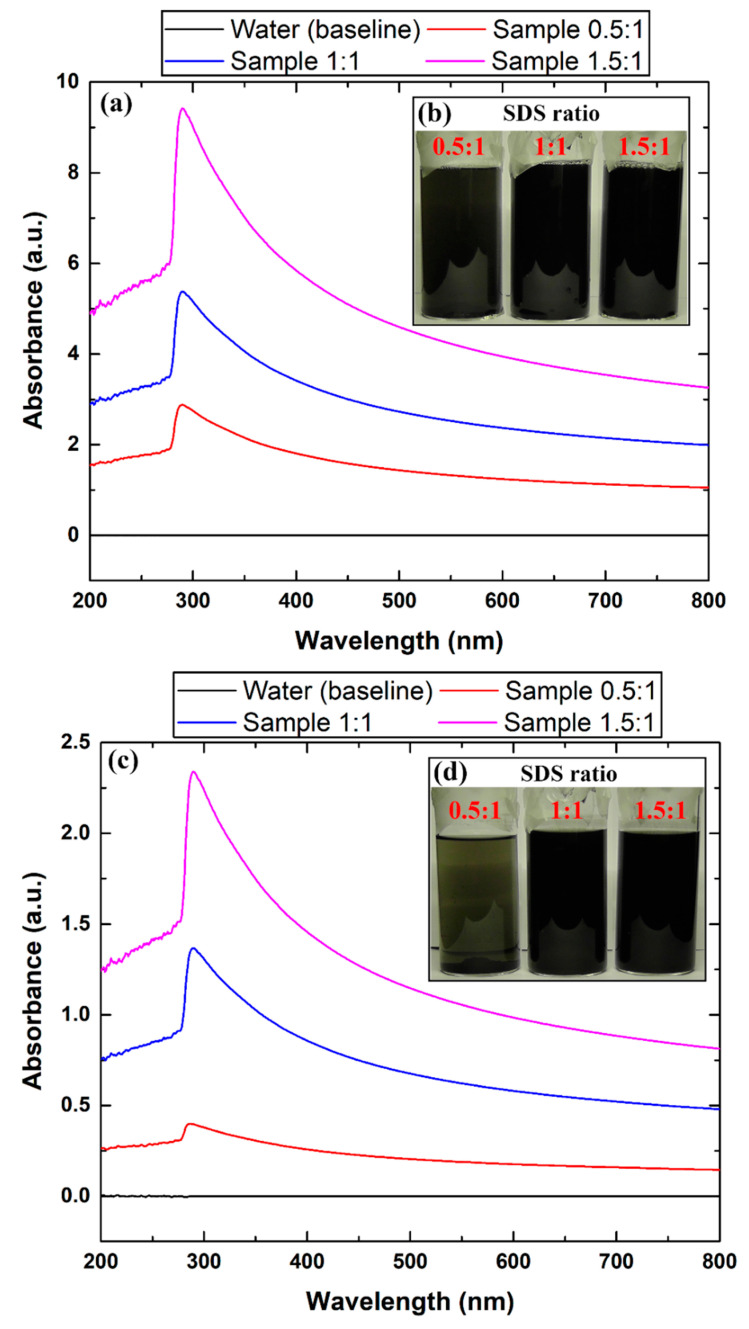
Dispersion stability of the 0.1 vol. % fabricated nanofluids at 30 °C with different SDS:graphene ratios, where (**a**,**b**) shows the UV–Vis and image capturing analysis directly after preparation, and (**c**,**d**) illustrate the same analysis on the 45th day.

## Data Availability

All data can be provided by the author upon request.

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
