# Peer review of "Graphene-Based Nanofluids: Production Parameter Effects on Thermophysical Properties and Dispersion Stability"

_nanomaterials, 2022, doi:10.3390/nano12030357_

Round 1

Reviewer 1 Report

Graphene-Based Nanofluids: Production Parameters Effect on Thermophysical Properties and Dispersion Stability

Comments:

  1. Abstract is too long; I think author should revise it carefully and make it more pruned.
  2. English language needs a substantial revision. Do not mix past and present tense.
  3. Author should more applications related to Graphene-Based Nanofluids instead of generalized nanofluids. It will be more appealing.
  4. In the conclusion, please show how the work advances the field from the present state of knowledge. Please provide a concise justification for your work in this section, indicating appropriate uses and extensions. Additionally, you can suggest new experiments/simulations and highlight those that are currently underway.
  5. Because there are so many symbols involved. Nomenclature has been added to help readers.
  6. The paper must be prepared according to the style of the journal.
  7. I found some self-citations: [1,6,7, 14,15,16,18]

Author Response

I would like to start by thanking the respected reviewer for accepting to review my manuscript and sharing his valuable time. I cordially acknowledge the useful comments and recommendations made by the respected reviewer on my manuscript. I have tried to revise the manuscript accordingly and the detailed corrections are listed below point by point. The respected reviewer can find all the modifications and corrections in the revised manuscript written in red text. The author respond to the respected reviewer queries are as follows:

Reviewer: Abstract is too long; I think author should revise it carefully and make it more pruned.

Author: I thank the respected reviewer for his suggestion, which I was more than happy to take into consideration. Kindly note that the abstract has been revised and improved as recommended. Thank you very much.

Reviewer: English language needs a substantial revision. Do not mix past and present tense.

Author: I thank the respected reviewer for his comment. Kindly note that the English language has been revised and enhanced throughout the manuscript. Thank you very much.

Reviewer: Author should more applications related to Graphene-Based Nanofluids instead of generalized nanofluids. It will be more appealing.

Author: I thank the respected reviewer for his suggestion, which was taken into account in the revised version of the manuscript. Kindly check page 2 and references [14-19]. Thank you very much.

Reviewer: In the conclusion, please show how the work advances the field from the present state of knowledge. Please provide a concise justification for your work in this section, indicating appropriate uses and extensions. Additionally, you can suggest new experiments/simulations and highlight those that are currently underway.

Author: I thank the respected reviewer for his comment. Kindly note that the conclusions section has been improved according to the respected reviewer recommendation. Thank you very much.

Reviewer: Because there are so many symbols involved. Nomenclature has been added to help readers

Author: I agree with the respected reviewer and apologies for not including the nomenclature. Kindly note that nomenclature has been added in the revised version of the manuscript as recommended by the respected reviewer. Thank you very much for pointing this out.

Reviewer: The paper must be prepared according to the style of the journal.

Author: I thank the respected reviewer for his comment and concern. Kindly note that once the manuscript gets accepted by the respected Editor, a member from the respected MDPI Nanomaterials staff will adjust the manuscript to the Journal style. This is usually done by them to provide the best possible illustration of article once published. Thank you very much.

Reviewer: I found some self-citations: [1,6,7, 14,15,16,18]  

Author: The respected reviewer is correct. However, these references are up to date and were used to support the scientific clams in which they were placed for. It also shows the readers the background of the author and contributes towards MDPI Nanomaterials and other MDPI respected journals since most of them have been published there. Thank you very much. 

Finally, the author would like to thank the respected reviewer for his time and very useful comments and remarks. Thank you very much.

Reviewer 2 Report

This paper is of interest to this journal, and the quality of this paper is fair. The presentation and methodology are good. I recommend its publication after considering the following points:

-The Abstract should be shortened.

- Add a nomenclature table and list all the parameters and abbreviations involved in the study. 

-The structure of this paper needs to be improved. The sections and the subsections in this paper should be renumbered.

- More physical insight of the Discussion section is needed. The author should ask the question "Why?"  in commenting on the results. Add more logical arguments besides the decreasing/increasing behavior of the profiles. 

- Some spelling checks should be corrected.

Author Response

I would like to start by thanking the respected reviewer for accepting to review my manuscript and sharing his valuable time. I cordially acknowledge the useful comments and recommendations made by the respected reviewer on my manuscript. I have tried to revise the manuscript accordingly and the detailed corrections are listed below point by point. The respected reviewer can find all the modifications and corrections in the revised manuscript written in red text. The author respond to the respected reviewer queries are as follows:

Reviewer: The Abstract should be shortened.

Author: I thank the respected reviewer for his suggestion, which I was more than happy to take into consideration. Kindly note that the abstract has been revised and improved as recommended. Thank you very much.

Reviewer: Add a nomenclature table and list all the parameters and abbreviations involved in the study.

Author: I thank the respected reviewer for his suggestion and apologies for not including the nomenclature from the beginning. Kindly note that nomenclature has been added in the revised version of the manuscript as recommended by the respected reviewer. Thank you very much for pointing this out.

Reviewer: The structure of this paper needs to be improved. The sections and the subsections in this paper should be renumbered.

Author: I thank the respected reviewer for his comment but I apologies because I am not sure which part of the manuscript you would like me to renumber. Kindly note that I tried to make the manuscript flow from the nanomaterial to the nanofluid. I hope that the respected reviewer will kindly accept this flow please. Thank you very much.     

Reviewer: More physical insight of the Discussion section is needed. The author should ask the question "Why?"  in commenting on the results. Add more logical arguments besides the decreasing/increasing behavior of the profiles.

Author: I thank the respected reviewer for his suggestion. Kindly note that his was taken into account in the revised version as best as possible. Kindly see page 15 and page 17. Thank you very much for pointing this out.

Reviewer: Some spelling checks should be corrected.

Author: I thank the respected reviewer for his comment. Kindly note that the English language has been revised and enhanced throughout the manuscript. Thank you very much.

Finally, the author would like to thank the respected reviewer for his time and very useful comments and remarks. Thank you very much.

Reviewer 3 Report

The present paper deals with ''Graphene-Based Nanofluids: Production Parameters Effect on Thermophysical Properties and Dispersion Stability''. I have some suggestions for the Authors that should be considered during the revision of the paper:

1. The general information should be removed in the abstract, even as background information. The abstract should be briefly written to describe the purpose of the research, the principal results, and major conclusions. Please revise. 

2. The originality and the novelty of the paper should be further justified.

3. The current Introduction should be improved. As for a good one, the authors should include at least four aspects: Background, Goals, Contribution/Innovation, and Organization of this study. 

4. I would like to recommend some newest related papers to enhance the research background of nanofluids:
https://doi.org/10.1016/j.jclepro.2020.124432

5. Please provide a good version of experimental test bench picture.

6. For the important data and equations, please add the references.

7. What methods for stability measurements are adopted? 

8. The authors are invited to add a paragraph discussing the limitation of this study. 

Author Response

I would like to start by thanking the respected reviewer for accepting to review my manuscript and sharing his valuable time. I cordially acknowledge the useful comments and recommendations made by the respected reviewer on my manuscript. I have tried to revise the manuscript accordingly and the detailed corrections are listed below point by point. The respected reviewer can find all the modifications and corrections in the revised manuscript written in red text. The author respond to the respected reviewer queries are as follows:

Reviewer: The general information should be removed in the abstract, even as background information. The abstract should be briefly written to describe the purpose of the research, the principal results, and major conclusions. Please revise.

Author: I thank the respected reviewer for his suggestion, which I was more than happy to take into consideration. Kindly note that the abstract has been revised and improved as recommended. Thank you very much.

Reviewer: The originality and the novelty of the paper should be further justified.

Author: I thank the respected reviewer for his recommendation. I have tried to further justify the novelty of the work as suggested in the revised version of the manuscript. Thank you very much.

Reviewer: The current Introduction should be improved. As for a good one, the authors should include at least four aspects: Background, Goals, Contribution/Innovation, and Organization of this study.

Author: I thank the respected reviewer for comment. I have tried to modify the manuscript according to the respected reviewer suggestion as best as possible. Kindly check the introduction section in the modified manuscript. Thank you very much.

Reviewer: I would like to recommend some newest related papers to enhance the research background of nanofluids:

https://doi.org/10.1016/j.jclepro.2020.124432.

Author: I thank the respected reviewer for his recommendation, which I found to be very useful to the introduction section of the manuscript. Kindly note that suggested literature was included in the revised version of the manuscript. Please check the red text in page 2. Thank you very much.

Reviewer: Please provide a good version of experimental test bench picture.

Author: I thank the respected reviewer for his suggestion. Kindly note that this is honestly the best image that I was able to get using my camera. I truly apologies for this and will try to purchase a new camera very soon with a set of lights to improve the quality of my images for future publications. Thank you very much.

Reviewer: For the important data and equations, please add the references.

Author: I thank the respected reviewer for his comment. Kindly note that references were added to the equations and to the data that were taken from the literature. Thank you very much.

Reviewer: What methods for stability measurements are adopted?

Author: I have used UV absorbance and image capturing techniques. I have included this in the revised version. Thank you very much.

Reviewer: The authors are invited to add a paragraph discussing the limitation of this study.

Author: I thank the respected reviewer for his invitation, which I happily took into account in the revised version of the manuscript. Kindly note that the limitation was added in the Conclusions section before the future research direction. Thank you very much for your suggestion.

Finally, the author would like to thank the respected reviewer for his time and very useful comments and remarks. Thank you very much.

Round 2

Reviewer 1 Report

IT can be acceptednow. 

Author Response

Thank you very much for your kindness and support.

Reviewer 3 Report

The paper has been well revised. I am glad to recommend it to be accepted.

Author Response

(The authors gave the same response as above.)
